# STACKING FOR TRANSFER LEARNING

## ABSTRACT

In machine learning tasks, overtting frequently crops up when the number of samples of target domain is insufficient, for the generalization ability of the classifier is poor in this circumstance. To solve this problem, transfer learning utilizes the knowledge of similar domains to improve the robustness of the learner. The main idea of existing transfer learning algorithms is to reduce the dierence between domains by sample selection or domain adaptation. However, no matter what transfer learning algorithm we use, the difference always exists and the hybrid training of source and target data leads to reducing fitting capability of the learner on target domain. Moreover, when the relatedness between domains is too low, negative transfer is more likely to occur. To tackle the problem, we proposed a two-phase transfer learning architecture based on ensemble learning, which uses the existing transfer learning algorithms to train the weak learners in the first stage, and uses the predictions of target data to train the final learner in the second stage. Under this architecture, the fitting capability and generalization capability can be guaranteed at the same time. We evaluated the proposed method on public datasets, which demonstrates the effectiveness and robustness of our proposed method.

## 1 INTRODUCTION

Transfer learning has attracted more and more attention since it was first proposed in 1995 Pan & Yang (2010) and is becoming an important field of machine learning. The main purpose of transfer learning is to solve the problem that the same distributed data is hard to get in practical applications by using different distributed data of similar domains. Several different kinds of transfer stratagies are proposed in recent years, transfer learning can be devided into 4 categories Weiss et al. (2016), including instance-based transfer learning, feature-based transfer learning, parameter-based transfer learning and relation-based transfer learning. In this paper, we focus on how to enhance the performance of instance-based transfer learning and feature-based transfer learning when limited labeled data from target domain can be obtained. In transfer learning tasks, when diff-distribution data is obtained to improve the generalization ability of learners, the fitting ability on target data set will be affected more or less, especially when the domains are not relative enough, negative transfer might occur Pan & Yang (2010), it's hard to trade off between generalization and fitting. Most of the existing methods to prevent negative transfer learning are based on similarity measure(*e.g.*, maximum mean distance(*MMD*), KL divergence), which is used for choosing useful knowledge on source domains. However, similarity and transferability are not equivalent concepts. To solve those problems, we proposed a novel transfer learning architecture to improve the fitting capability of final learner on target domain and the generalization capability is provided by weak learners. As shown in Figure 1, to decrease the learning error on target training set when limited labeled data on target domain can be obtained, ensemble learning is introduced and the performances of transfer learning algorithms are significantly improved as a result.

In the first stage, traditional transfer learning algorithms are applied to diversify training data(*e.g.*, Adaptive weight adjustment of boosting-based transfer learning or different parameter settings of domain adaptation). Then diversified training data is fed to several weak classifiers to improve the generalization ability on target data. To guarantee the fitting capability on target data, the predictions of target data is vectorized to be fed to the final estimator. This architecture brings the following advantages:

- When the similarity between domains is low, the final estimator can still achieve good performance on target training set.

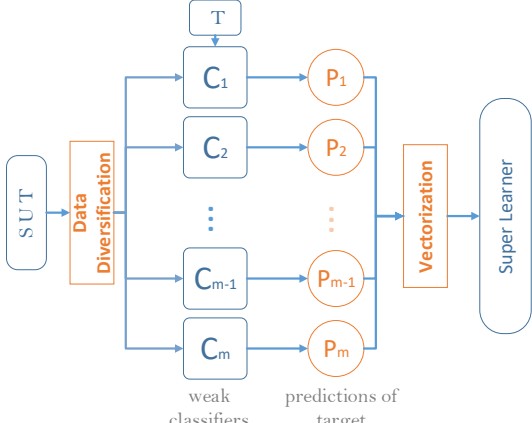

Figure 1: Overview of stacking for transfer learning. Firstly, source data and target data are hybrid together to train the weak learners, then super learner is used to fit the predictions of target data.

- Parameter setting is simplified and performance is better than individual estimators under normal conditions.

To test the effectiveness of the method, we respectively modified *TrAdaboost* Dai et al. (2007) and *BDA* Wang et al. (2017) as the base algorithms for data diversification and desired result is achieved.

## 1.1 RELATED WORK

### 1.1.1 INSTANCE-BASED TRANSFER LEARNING

*TrAdaboost* proposed by Dai et al. (2007) is a typical instance-based transfer learning algorithm, which transfer knowledge by reweighting samples of target domain and source domain according to the training error. In this method, source samples are used directly for hybrid training. As the earliest boosting based transfer learning method, there are many inherent defects in *TrAdaboost* (*e.g.*, high requirements for similarity between domains, negative transfer can easily happen). Moreover, *TrAdaboost* is extended from *Adaboost* and use *WMA*(Weighted Majority Algorithm) Littlestone & Warmuth (1994) to update the weights, and the source instance that is not correctly classified on a consistent basis would converge to zero by $\lceil N/2 \rceil$ and would not be used in the final classifier's output since that classifier only uses boosting iterations$\lceil N/2 \rceil \rightarrow N$. Two weakness caused by disregarding first half of ensembles analysised in Al-Stouhi & Reddy (2011) are in the list below:

- As the source weights convergence rapidly, after $\lceil N/2 \rceil$ iterations, the source weights will be too low to make full use of source knowledge.
- Later classifiers merely focus on the harder instances.

To deal with rapid convergence of *TrAdaboost*, Eaton & Desjardins (2009) proposed *TransferBoost*, which apply a 2-phase training process at each iteration to test whether negative transfer has occurred and adjust the weights according to the results. Al-Stouhi & Reddy (2011) introduces an adaptive factor in weights update to slow down the convergence. Yao & Doretto (2010) proposed multi-source *TrAdaboost*, aimed at utilize instances from multiple source domains to improve the transfer performance. In this paper, we still use the *WMA* to achieve data diversification in experiment of instance-based transfer learning, but stacking rather than boosting is used in final predictions.

### 1.1.2 FEATURE-BASED TRANSFER LEARNING

Feature based transfer learning mainly realizes transfer learning by reducing the distribution difference(*e.g.*, *MMD*) between source domain and target domain by feature mapping, which is the most studied method for transfer knowledge in recent years.

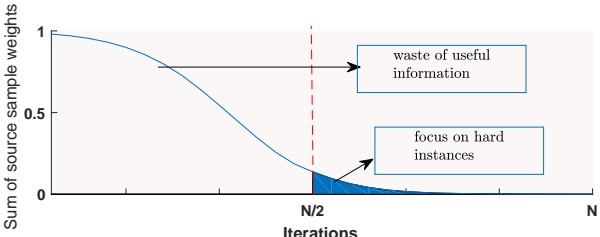

Figure 2: Weights of the source samples and target samples change with the iterations in *TrAdaboost*. All of the weights are initialized to 1.

Pan et al. (2011) proposed transfer components analysis(*TCA*) as early as 2011, *TCA* achieve knowledge transfer by mapping the feature of source domain and target domain to a new feature space where the *MMD* between domains can be minimized. However, not using labels brings a defect that only marginal distribution can be matched. To address the problem, Long et al. (2014) proposed joint distribution adaptation(*JDA*) which fit the marginal distribution and conditional distribution at the same time, for unlabeled target data, it utilizes pseudo-labels provided by classifier trained on source data. After that, Wang et al. (2017) extended *JDA* for imbalanced data.

In neural networks, it's easy to transfer knowledge by pre-train and fine-tune because feature extracted by lower layers are mostly common for different tasks Yosinski et al. (2014), to transfer knowledge in higher layers which extract task-specific features, Tzeng et al. (2014), Long et al. (2015) and Long et al. (2016) add *MMD* to the optimization target in higher layers.

### 1.1.3 NEGATIVE TRANSFER

The learning performance in the target domain using the source data could be poorer than that without using the source data, This phenomenon is called negative transfer Pan & Yang (2010). To avoid negative transfer, Liu et al. (2016) point out that one of the most important research issues in transfer learning is to determine whether a given source domain is effective in transferring knowledge to a target domain, and then to determine how much of the knowledge should be transferred from a source domain to a target domain. Researchers have proposed to evaluate the relatedness between the source and target domains. When limited labeled target data can be obtained, two of the methods are listed below:

- Introduce the predened parameters to qualify the relevance between the source and target domains. However, it is very labor consuming and time costing to manually select their proper values.
- Examine the relatedness between domains directly to guide transfer learning.

The notion of positive transferability was first introduced in Seah et al. (2013) for the assessment of synergy between the source and target domains in their prediction models, and a criterion to measure the positive transferability between sample pairs of different domains in terms of their prediction distributions is proposed in that research. Shao (2016) proposed a kernel method to evaluate the task relatedness and the instance similarities to avoid negative transfer Gui et al. (2017) proposed a method to detection the occurance of negative transfer which can also delay the point of negative transfer in the process of transfer learning. Ge et al. (2014) remind that most previous work treats knowledge from every source domain as a valuable contribution to the task on the target domain could increase the risk of negative transfer. A two-phase multiple source transfer framework is proposed, which can effectively downgrade the contributions of irrelevant source domains and properly evaluate the importance of source domains even when the class distributions are imbalanced.

### 1.2 STACKED GENERALIZATION

Stacking is one of the ensemble learning methods that fuses multiple weak classifiers to get a better performance than any single one Kai & Witten (1997). When using stacking, diversification of weak

learners has an important impact on the performance of ensemble Woniak et al. (2014). Here are some common ways to achieve diversification:

- **Diversifying input data**: using different subset of samples or features.
- **Diversifying outputs**: classifiers are only for certain categories.
- **Diversifying models**: using different classification models.

In this paper, we can also regard the proposed architecture as a stacking model which uses transfer learning algorithms to achieve input diversification.

## 2 STACKING FOR TRANSFER LEARNING

### 2.1 STACKING FOR INSTANCE-BASED TRANSFER LEARNING

In this section, we introduce how the instance-based transfer learning is applied to the proposed architecture. we use *TrAdaboost* as an example and make a simple modification to turn it to stacking. In *TrAdaboost*, we need a large number of labeled data on source domain and limited labeled data on the target domain. We use $X = X_S \cup X_T$ to represent the feature space. Source space($X_S$) and target space($X_T$) are defined as $x_i^S \in X_S(i = 1, ..., n)$ and $x_j^T \in X_T(j = 1, ..., m)$ respectively. Then the hybrid training data set is defined in equation 1.

$$x_i = \begin{cases} x_i^S, & i = 1, 2, ..., n; \\ x_{i-n}^T, & i = n+1, n+2..., n+m. \end{cases} \tag{1}$$

Weight vector is initialized firstly by $\mathbf{w}^1 = (w_1^1, w_2^1, ..., w_{n+m}^1)$, in the $t^{th}$ iterration, let $\mathbf{w}^t$ be the sample weight when training the weak classifier and use Equation 2 to calclulate the weighted error rate.

$$\epsilon_t = \sum_{i=1+n}^{m+n} \frac{w_i^t \cdot \mathbb{I}(h_t(x_i) \neq c(x_i))}{\sum_{i=n+1}^{n+m} w_i^t} \tag{2}$$

In $t_{th}$ iteration, the weights are updated by Equation 3.

$$w_i^{t+1} = \begin{cases} w_i^t \beta^{|h_t(x_i)-c(x_i)|}, & 1 \leq i \leq n; \\ w_i^t \beta_t^{-|h_t(x_i)-c(x_i)|}, & n+1 \leq i \leq n+m. \end{cases} \tag{3}$$

Here, $\beta_t = \frac{\epsilon_t}{1-\epsilon_t}$ and $\beta = 1/(1+\sqrt{2\ln n/N})$. It is noteworthy that original *TrAdaboost* is for binary classification. In order to facilitate experimental analysis and comparison, we extend the traditional *TrAdaboost* to a multi-classification algorithm according to *Multi-class Adaboost* proposed in Zhu et al. (2009), then $\beta_t$ and $beta$ defined as:

$$\begin{aligned} \beta_t &= \frac{\epsilon_t}{(1-\epsilon_t)(K-1)} \\ \beta &= \frac{1}{(1+\sqrt{2\ln n/N})(K-1)} \end{aligned} \tag{4}$$

K is the class number. Equation 5 defines the final output for each class.

$$P^k(x_i) = \prod_{t=\lceil N/2 \rceil}^{N} \beta_t^{h_t^k(x_i)} \tag{5}$$

Moreover, for single-label problem, we use softmax to transfer $P^k(x)$ to probabilities. To address the rapid convergence problem, Eaton & Desjardins (2009) proposed *TransferBoost*, which utilizes all the weak classifiers for ensemble, but in the experiment, early stop can improve the final performance, so which classifiers should be chosen is still a problem. Al-Stouhi & Reddy (2011) proposed *dynamic TrAdaboost*, In this algorithm, an adaptive factor is introduced to limit the convergence rate of source sample weight. However, it's not always effective in practical use. Theoretical upper

bound of training error in target domain is not changed in *dynamic TrAdaboost*, which is related to the training error on target domain, we have:

$$\epsilon \leq 2^{\lceil N/2 \rceil} \prod_{t=\lceil N/2 \rceil}^{N} \sqrt{\epsilon_t(1-\epsilon_t)} \tag{6}$$

---

**Algorithm 1** stacked generalization for instance-based transfer learning.

1: **input** labeled data sets $D_s$ and $D_t$, source task $S = \{(x_i^S, c_i)\}_{i=1}^n$ and target task $T = \{(x_i^T, c_i)\}_{i=1}^m$.
2: **initialize** the weight vector $\mathbf{w}^1 = (w_1^1, w_2^1, ..., w_{n+m}^1)$ and $\lambda$.
3: **for** each $t$ from 1 to N **do**
4:     Call Learner, providing it labeled target data set with the distribution $\mathbf{w}^t$. Then get back a hypothesis of S.
5:     Calculate the error on S:
6:
$$\epsilon_t^i = \frac{w_i^t \mathbb{I}(h_t(x_i) \neq c(x_i))}{\sum_{i=1}^n w_i^t}$$

7:     Get a subset of source task $S_t = \{(x_i, c_i), \epsilon_t^i < \lambda\}_{i=1}^n$.
8:     Call learner, providing it $S_t \cup T$. Then get back a hypothesis of T.
9:     Calculate the error on T using equation. 2.
10:     updata $\beta_t$ using equation. 4.
11:     Update the new weight vector using equation. 3.
12: **end for**
13: Construct probability vectors by concatenating

$$\mathbf{O}_{t,i} = (softmax(P_t^1(x_i)), ..., softmax(P_t^K(x_i)))$$

.
14: Construct target feature matrix $\mathbf{Z}$, $\mathbf{Z} \in \mathbb{R}^{m \times (NK)}$, $\mathbf{Z}_i = (\mathbf{O}_{1,i}, ..., \mathbf{O}_{N,i})$.
15: Train final learner providing $\{(\mathbf{Z}_i, c_i)\}_{i=n+1}^{n+m}$.

---

Although dynamic *TrAdaboost* can improve the weights of source samples after iteration $\lceil N/2 \rceil$ and the generalization capability is improved, it's very likely that the error rate on source domain $\epsilon_t$ increases, sometimes it even aggravates the occurrence of negative transfer when the domains are not similar enough. We use stacking to address the problems above in this section, in the data diversification stage, *TrAdaboost* is used to reweight samples for each weak classifier. Meanwhile, because we make use of all the weak classifiers, to avoid the high source weights of irrelative source samples negatively effects on the task in early iterations, a two-phase sampling mechanism is introduced in our algorithm.

A formal description of stacking for instance-based transfer learning is given in *Algorithm 1*. The main difference between stacking for instance-based transfer learning and *TrAdaboost* are listed as follows:

- A two-phase sampling process and an extra parameter $\lambda$ is introduced. Firstly, target data is fed to weak learner and the weighted error rate of source samples are used to decide which samples can be used for hybrid learning by comparing with the threshold $\lambda$. As the source weights reduces with the number of iterations increasing, more and more source samples will be utilized.

- Stacking rather than TrAdaboost is used to get the final output. We construct a feature matrix by the outputs of weak learners on target data, then use a super learner(*e.g.*, LogitRegression in our experiment) to fit the labels. In this way, training error on target set can be minimized.

When compared with *TrAdaboost*, stacking is insensitive to the performance of each weak classifier because the training error on target data can be minimized in stacking, which means it's more robust in most cases and brings some benefits:

- When using stacking, all of the weak classifiers could be used.
- When source domain is not related enough, stacking performs better.

## 2.2 STACKING FOR FEATURE-BASED TRANSFER LEARNING

One of a popular methods for feature-based transfer learning to achieve knowledge transfer is domain adaptation, which minimizes the distribution difference between domains by mapping features to a new space, where we could measure the distribution difference by *MMD*. Generally speaking, we use $P(X_S)$, $P(X_T)$ and $P(Y_S|X_S)$, $P(Y_T|X_T)$ to represent the marginal distribution and conditional distribution of source domain and target domain respectively. In Pan et al. (2011), transfer component analysis(*TCA*) was proposed to find a mapping which makes $P(\phi(X_S)) \approx P(\phi(X_T))$, the *MMD* between domains in *TCA* is defined as:

$$dist(D_X, D_T) = \|\frac{1}{n_1}\sum_{i=1}^{n_1}\phi(x_i^S) - \frac{1}{n_2}\sum_{i=1}^{n_2}\phi(x_i^T)\| \tag{7}$$

Long et al. (2014) proposed joint distibution adaptation(*JDA*) to minimize the differences of marginal distribution and conditinal distribution at the same time, the *MMD* of conditional distribution is defined as:

$$dist(X_S, X_T) = \sum_{c=1}^{C}\|\frac{1}{n_c}\sum_{x_i^S \in X_S^{(c)}}\phi(x_i^S) - \frac{1}{m_c}\sum_{x_i^T \in X_T^{(c)}}\phi(x_i^T)\| \tag{8}$$

In Wang et al. (2017), balanced distribution adaptation was proposed, in which algorithm, an extra parameter is introduced to adjust the importance of the distributions, the optimization target is defined by Equation 9:

$$dist(X_S, X_T) = \mu\|\frac{1}{n_1}\sum_{i=1}^{n_1}\phi(x_i^S) - \frac{1}{n_2}\sum_{i=1}^{n_2}\phi(x_i^T)\| +$$

$$(1-\mu)\sum_{c=1}^{C}\|\frac{1}{n_c}\sum_{x_i^S \in X_S^{(c)}}\phi(x_i^S) - \frac{1}{m_c}\sum_{x_i^T \in X_T^{(c)}}\phi(x_i^T)\| \tag{9}$$

To solve the nonlinear problem, we could use a kernel matrix defined by: $\mathbf{K} = \begin{bmatrix} \mathbf{K}_{src,src} & \mathbf{K}_{src,tar} \\ \mathbf{K}_{tar,src} & \mathbf{K}_{tar,tar} \end{bmatrix}$, then the optimization proplem can be formalized as:

$$\min_{\mathbf{A}} tr\left(\mathbf{A}^\top\mathbf{K}\left((1-\mu)\mathbf{M}_0 + \mu\sum_{c=1}^{C}\mathbf{M}_c\right)\mathbf{K}^\top\mathbf{A}\right) + \lambda\|\mathbf{A}\|_F^2$$

$$s.t. \mathbf{A}^\top\mathbf{K}\mathbf{H}\mathbf{K}^\top\mathbf{A} = \mathbf{I}, 0 \le \mu \le 1 \tag{10}$$

Where $\mathbf{H}$ is a centering matrix, $\mathbf{M}_0$ and $\mathbf{M}_c$ represent the *MMD* matrix of marginal distribution and conditional distribution respectively, A is the mapping matrix.

In domain adaptation, performace is sensitive to the selection of parameters(*e.g.*, kernel type, kernel param or regularization param). For instance, if we use $rbf$ kernel, as defined in Equation 11, to construct the kernel matrix. selection of kernel param $\sigma$ has an influence on the mapping. In this paper, we use *BDA* as a base algorithm in the proposed architecture to achieve data diversification by using different kernel types and parameter settings.

$$\kappa(x, y) = e^{(-\frac{\|x-y\|^2}{2\sigma^2})} \tag{11}$$

By taking adavantage of stacking, we could get a better transfer performance than any single algorithm. Here, we introduce how we choose the kernel parameter in our experiments. In ensemble learning, it's significant to use unrelated weak classifiers for a better performance Woniak et al.

(2014)(*i.e.*, learners should have different kinds of classification capabilities). moreover, performances of learners shouldn't be too different or the poor learners will have an negative effect on ensemble. In another word, we choose the kernel parameter in a largest range where the performance is acceptable. We take the following steps to select parameters. Firstly, search a best value of kernel parameter $\sigma$ for weak classifier, where the accuracy on validation set is $Accuracy_{max}$. secondly, set a threshold parameter $\lambda$ and find an interval $(\sigma_{min}, \sigma_{max})$ around $\sigma$ where the accuracy on validation set satisfy $Accuracy_{max} - \lambda \leq Accuracy$ when $\sigma \in (\sigma_{min}, \sigma_{max})$. Finally, select N parameters in $(\sigma_{min}, \sigma_{max})$ by uniformly-spaced sampling. When multiple type of kernels are utilized, we choose parameter sets for each seperately by repeating the above steps. In our method, the settings of $\lambda$ and N should be taken into consideration, if $\lambda$ is too large, the performance of each learner can't be guaranteed, if $\lambda$ is too small, training data can't be diversified enough. Set N to a large number would help to get a better performance in most cases, while the complexity could be high.

---

**Algorithm 2** stacked generalization for feature-based transfer learning.

1: **input** Source and target feature maxtrix $\mathbf{X}_{src}, \mathbf{X}_{tar}$, label vector $\mathbf{Y}_{src}, \mathbf{Y}_{tar}$.
2: **Initialize** Regularization parameter $\lambda$, kernel functions $\{\kappa_t\}_{t=1}^N$, number of classes $K$, $\mathbf{M}_0$ and $\mathbf{M}_c$.
3: **for** each $t$ from 1 to N **do**
4:     Construct $\mathbf{X} = \begin{bmatrix} \mathbf{X}_{src} \\ \mathbf{X}_{tar} \end{bmatrix}, \mathbf{Y} = \begin{bmatrix} \mathbf{Y}_{src} \\ \mathbf{Y}_{tar} \end{bmatrix}$
5:     Contruct kernel matrix $\mathbf{K}_t$ using $\kappa_t$.
6:     Solve the eigendecomposition problem and use d smallest eigenvectors to build $\mathbf{A}_t$.
7:     Train $t_{th}$ leaner on $\{\mathbf{A}_t^\top \mathbf{K}_t, \mathbf{Y}\}$.
8:     Call learner t providing $\mathbf{A}_t^\top \mathbf{K}_t^{tar}$, Get hypothesis $h_t$.
9:     Construct probability vectors by concatenating $\mathbf{O}_{t,i} = (h_{t,i}^1, ..., h_{t,i}^K)$, where K is the number of classes.
10: **end for**
11: Construct target feature matrix $\mathbf{Z}, \mathbf{Z} \in \mathbb{R}^{m \times (NK)}$.$\mathbf{Z}_i = (\mathbf{O}_{1,i}, ..., \mathbf{O}_{N,i})$.
12: Train final learner providing $(\mathbf{Z}, \mathbf{Y}_{tar})$.

---

*Algorithm 2* presents the detail of our method. In our algorithm, kernel function $\kappa_t$ can be differentiated by kernel types or kernel params. In $t^{th}$ iteration, we choose $\kappa_t$ to mapping the feature space, then get the matrix $\mathbf{A}$ by *BDA*, $\mathbf{A}^\top \mathbf{K}^{tar}$ and $\mathbf{A}^\top \mathbf{K}^{src}$ are feed to weak learner. After that, we concatenate predictions of $\mathbf{A}^\top \mathbf{K}^{tar}$ as features of the super leaner. In this paper, we assume that there are limited labeled data on target set, so we use modified *BDA*, which uses real label rather than pseudo label, to adapt conditional distribution.

## 3 EXPERIMENT SET UP

### 3.1 DATA SET

To evaluate the effectiveness of our method, 6 public datasets and 11 domains as shown in Table 1 are used in our experiment of instance-based transfer learning and feature-based transfer learning. The detail is described as follow:

Table 1: 6 datasets(11 domains) for experiment.

| *Dataset* | *Domain* | #samples | #classes |
|---|---|---|---|
| mnist | M | 5000 | 10 |
| usps | U | 5000 | 10 |
| 20newsgroup | rec+talk+sci | - | 2 |
| COIL20 | COIL1+COIL2 | 720 + 720 | 20 |
| Caltech | C | 1123 | 10 |
| office | A+W+D | 985+295+157 | 10 |

- *mnist* and *USPS* are standard digit recognition datasets, we select 5000 of each and resize usps to 28*28 for our experiment.

- *rec vs. talk*, *rec vs. sci* and *sci vs. talk* are from *20newsgroups*, which has a hierarchical structure, we construct source domain and target domain by combining different sub-categories for instance-based transfer learning, the method is described in Dai et al. (2007).

- *COIL20* is an image recognition dataset contains 20 classes, which can be divided into *COIL2* and *COIL1*, *COIL2* is obtained after 90 degrees of rotation from COIL1.

- *Caltech(C)*, *amazon(A)*, *webcam(W)*, *dslr(D)* are four image recognition domains, we choose 10 of the categories to conduct transfer learning between them.

### 3.1.1 EXPERIMENT RESULT

### 3.1.2 INSTANCE BASED TRANSFER

We selected *mnist*, *USPS*, *20newsgroups* and *COIL20* datasets for instance-based transfer learning. Experiment is carried out under different iterations and different ratios of $\#source$ and $\#target$ in training set. To verify the effectiveness of the proposed algorithm, we compared our stacking method for transfer learning with *TrAdaboost* and choose *Randomforest* as the weak learners, the super learner of stacking is *LogitRegression*.

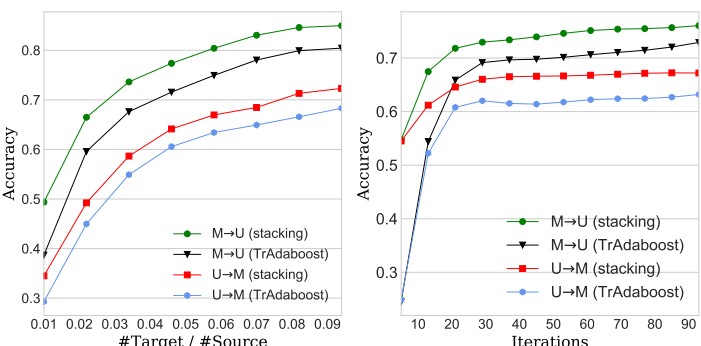

Figure 3: Accuracy of instance-based transfer on mnist vs. usps changed with the iterations and ratio of $\#target$.

Figure 3 shows the experiment results of transfer between mnist and *USPS* under different iterations and ratios of $\#source$ and $\#target$. We observe that stacking achieves much better performance than *TrAdaboost*. Firstly, acuuracy of stacking method is higher when ratio changing from **1%** to **10%**. Especially, the fewer the labeled target samples are, the more improvement stacking method could achieve. Secondly, few iterations are required for stacking method to achieve a relatively good performance, when the curve is close to convergence, there's still about **5%** improvement compared with *TrAdaboost*. Moreover, in both transfer tasks *USPS→ mnist* and *mnist → USPS*, stacking method performs significantly better than *TrAdaboost*. The reason why stacking performs better is analyzed in section 3.1, we assume that the introduction of source leads to under fitting on target data when hybrid training, to confirm our hypothesis, we made a comparision of training error on source data and target data between *TrAdaboost* and stacking method, Figure 4 shows the result of four of the transfer tasks.

Table 2 shows the results of all the transfer tasks under 20 iterations.

### 3.1.3 FEATURE BASED TRANSFER

*BDA* is chosen as the base algorithm to achieve data diversification in our experiment, we mainly test the infulence of different kernel functions has on the perfomance and the effectiveness of the method proposed in section 3.2. The ratio of $\#source$ and $\#target$ is set to **5%** and we use *rbf* kernel,

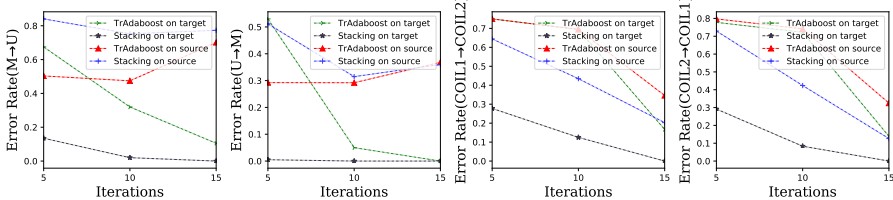

Figure 4: Training error on source and target data changed with iterations in transfer tasks $USPS \rightarrow$ $mnist$, $mnist \rightarrow USPS$, $COIL1 \rightarrow COIL2$ and $COIL2 \rightarrow COIL1$.

Table 2: Comparison between *TrAdaboost* and stacking under different ratios of $\#target$.

| Transfer task | TrAdaboost | | | | Stacked Generalization | | | |
|---|---|---|---|---|---|---|---|---|
| | 1% | 5% | 10% | 15% | 1% | 5% | 10% | 15% |
| $mnist \rightarrow usps$ | 0.3864 | 0.7260 | 0.8086 | 0.8280 | **0.4826** | **0.7878** | **0.8500** | **0.8672** |
| $usps \rightarrow mnist$ | 0.2826 | 0.6066 | 0.6854 | 0.7337 | **0.3352** | **0.6486** | **0.7231** | **0.782** |
| $COIL1 \rightarrow COIL2$ | - | 0.3610 | 0.4331 | 0.4462 | - | **0.5352** | **0.5401** | **0.6372** |
| $COIL2 \rightarrow COIL1$ | - | 0.3932 | 0.4521 | 0.4428 | - | **0.5555** | **0.5586** | **0.6111** |
| $rec \rightarrow talk$ | - | 0.7512 | 0.7908 | 0.8627 | - | **0.8465** | **0.8845** | **0.9176** |
| $rec \rightarrow sci$ | - | 0.5434 | 0.7087 | 0.7683 | - | **0.6980** | **0.7609** | **0.8307** |
| $sci \rightarrow talks$ | - | 0.7230 | 0.7177 | 0.7156 | - | **0.7583** | **0.7747** | **0.7747** |

*poly* kernel and *sam* kernel to conduct our experiment, for the sake of simplicity, kernel function is defined in Table 3, where $\gamma$ is variable for different weak learners.

Table 3: Definitions of kernel functions.

| rbf | $\kappa(x, y) = e^{-\gamma||x-y||^2}$ |
|---|---|
| sam | $\kappa(x, y) = e^{-\gamma arccos(x^\mathrm{T} y)^2}$ |
| poly | $\kappa(x, y) = (x^\mathrm{T} y)^\gamma$ |

To observe how the selection of kernels affects the feature distribution, we visualize the feature representations learned by *BDA* under different kernel parameters and kernel types when adapting domains *A* and *C*. As shown in Figure 5(a), data distribution and similarity between domains change with the kernel parameters, when compared with Figure 5(b), which presents feature distribution of $rbf$ kernel, it's obvious that using different kernel types can provide more diversity.

In this paper, to construct the kernel set by sampling parameters in a range where the performance is not too worse than the best one, we followed the method given in section 3.2 and set threshold $\lambda$ varies from **5%** to **10%** for different tasks. For each kernel type, we select *10* different parameters(*i.e.*, **10** weak classifiers) for stacking.

Tabel 4 shows the comparison between single algorithm and ensemble learning. For each kernel type, we give the best accuracy, average accuracy of weak learners and accuracy of ensemble learning, *Randomforest* and *LogitRegression* as the weak learner and super learner respectively. Accuracy of integrating all the kernel types is shown in the last column and the best performance of each task is in bold. We can learn from the table that ensemble learning outperfoms the best single learner in all the tasks, and in most cases, using both of $rbf$ and another type of kernels are able to improve the performance. However, when should we use multiple kernel types in stacking needs to be further studied.

In summary, the reason why the proposed method can improve the performance of feature-based transfer learning is listed as follows:

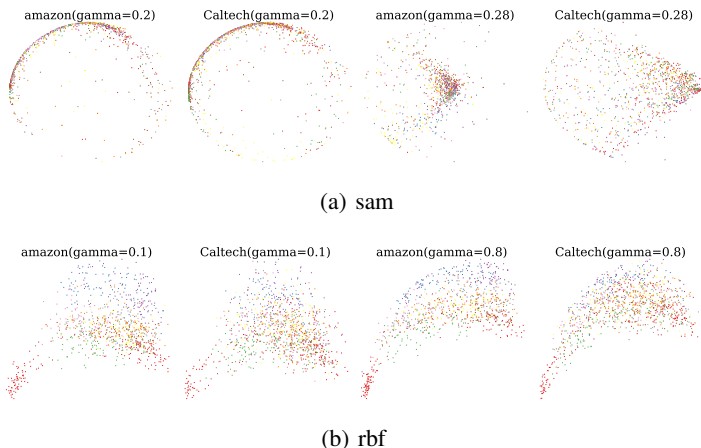

Figure 5: Feature visualization of the A→C task under different kernels.

Table 4: Performance of Feature-based Transfer.

| Transfer task | rbf | | | sam/poly | | | stacking |
|---|---|---|---|---|---|---|---|
| | avg | best | stacking | avg | best | stacking | |
| $mnist \rightarrow usps$ | 0.461 | 0.514 | 0.548 | 0.423 | 0.459 | 0.557 | **0.604** |
| $uspt \rightarrow mnist$ | 0.527 | 0.607 | 0.692 | 0.567 | 0.619 | 0.716 | **0.727** |
| $COIL1 \rightarrow COIL2$ | 0.913 | 0.956 | **0.987** | 0.811 | 0.842 | 0.902 | 0.983 |
| $COIL2 \rightarrow COIL1$ | 0.882 | 0.939 | 0.948 | 0.798 | 0.833 | 0.897 | **0.963** |
| $A \rightarrow C$ | 0.420 | 0.427 | **0.439** | 0.411 | 0.420 | 0.424 | **0.439** |
| $A \rightarrow D$ | 0.503 | 0.514 | **0.551** | 0.482 | 0.495 | 0.523 | **0.551** |
| $A \rightarrow W$ | 0.401 | 0.445 | 0.473 | 0.465 | 0.486 | **0.522** | 0.497 |
| $C \rightarrow A$ | 0.478 | 0.500 | 0.570 | 0.503 | 0.520 | 0.561 | **0.577** |
| $C \rightarrow D$ | 0.485 | 0.495 | 0.505 | 0.476 | 0.504 | **0.514** | 0.504 |
| $C \rightarrow W$ | 0.530 | 0.546 | **0.588** | 0.554 | 0.563 | 0.570 | **0.588** |
| $D \rightarrow A$ | 0.395 | 0.417 | 0.450 | 0.395 | 0.471 | 0.481 | **0.494** |
| $D \rightarrow C$ | 0.325 | 0.336 | 0.372 | 0.337 | 0.353 | 0.360 | **0.386** |
| $W \rightarrow A$ | 0.405 | 0.433 | 0.446 | 0.427 | 0.450 | 0.464 | **0.472** |
| $W \rightarrow C$ | 0.319 | 0.339 | 0.352 | 0.327 | 0.339 | 0.358 | **0.362** |

Firstly, we use super learner to fit the target domain, so the bias of weak learners introduced by hybrid training with source data is reduced.

Secondly, multiple kernels are utilized to achieve data diversification, so we could integrate the classification ability of weak learners trained on diff-distribution data.

## 4 CONCLUSION

In this paper, we proposed a 2-phase transfer learning architecture, which uses the traditional transfer learning algorithm to achieve data diversification in the first stage and the target data is fitted in the second stage by stacking method, so the generalization ability and fitting ability on target data could be satisfied at the same time. The experiment of instance-based transfer learning and feature-based transfer learning on 11 domains proves the validity of our method. In summary, this framework has the following advantages:

- No matter if source domain and target domain are similar, the training error on target data set can be minimized theoretically.
- We reduce the risk of negative transfer in a simple and effective way without a similarity measure.

- Introduction of ensemble learning gives a better performance than any single learner.
- Most existing transfer learning algorithm can be integrated into this framework.

Moreover, there're still some problems require our further study, some other data diversification method for transfer learning might be useful in our model, such as changing the parameter $\mu$ in *BDA*, integrating multiple kinds of transfer learning algorithms, or even applying this framework for multi-source transfer learning.

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
