# OpenReview forum: "Stacking for Transfer Learning"
_ICLR.cc/2019/Conference_

### Official Review · AnonReviewer2 · 2018-11-01
**No technical contributions, a lot of typos and grammar errors**

**Rating:** 2
**Confidence:** 5

**Review:**

In this paper, the authors proposed to learn a stacked classifier on top of the outputs of well-known transfer learning models for transfer learning. The authors claimed that their proposed solution can avoid negative transfer.

Technically, there are no contributions. The proposed solution is a straight-forward A+B, where both A and B are well-known. Specifically, in the proposed solution, different well-known transfer learning models are used as the 1st level classifiers to generate intermediate outputs, then a stacked classifier is trained with the intermediate outputs as its inputs. Stacking techniques are also well-known in ensemble learning. Therefore, I do not see any new technical ideas.

Moreover, the proposed solution cannot really avoid negative transfer. If two domains are indeed very different, the performance of the basic transfer learning models would be very bad, e.g., worse than random guess. In this case building a stacked classifier cannot help to boost the final performance.

The datasets used to conduct experiments are all of toy sizes.

There are a lot of typos and grammar errors. The format of citations in the main text are incorrect.

In summary, the quality of this paper is far below the standard of top conferences.

---

### Official Review · AnonReviewer1 · 2018-11-01
**Recommend rejection of the paper due to the limited contributions and preliminary experiments.**

**Rating:** 4
**Confidence:** 5

**Review:**

The authors proposed a stacking method for both instance-based and feature-based transfer learning based on a two-phase strategy. It first introduced some self-defined parameters to diversify the data or the model, and then adopted some existing transfer learning to train the model.

Strength:
1) A stacking method for both instance-based and feature-based transfer learning.

Weakness:
1) Incremental contributions and limited novelty.
2) Some claims are not well supported.
3) Experimental results are preliminary.

The technical contribution of this work is limited. The main difference between the proposed instance-based stacking method and TrAdaboost are twofold: 1) using a self-defined threshold to select a subset of source samples for training; 2) using stacking instead of TrAdaboost to get the final output. The improvements upon TrAdaboost are marginal. Also, for the proposed feature-based stacking method, it just used the different kernel parameter values to diversify the model. The novelty is trivial.

Several claims in the paper are not well discussed and/or evidence-supported, such as：
1) “When the similarity between domains is low, the final estimator can still achieve good performance on target training set.”
2) “When source domain is not related enough, stacking performs better.”
3) “We reduce the risk of negative transfer in a simple and effective way without a similarity measure.”
More discussions should be given, for example, how to measure the domain similarity and how to reduce the risk of negative transfer. Also, there are no experimental results to support the claims.

For the evaluation, it is inappropriate to choose Randomforest as the weak leaners since Randomforest is an ensemble method. It is better to give some explanation on how the data distribution and similarity between domains change with the kernel parameters in Figure 5.

For the algorithm comparison, only TrAdaboost is used as the baseline to compare with the proposed instance-based stacking method. The results could be more convincing if some recent ensemble-based transfer learning methods are included for comparison. For the evaluation of the proposed feature-based stacking method, the authors should at least compare their method with BDA since BDA is used as its base algorithm.

Some symbols used in equations are not defined, such as h_t and c in Equation (2).

The paper needs a careful proofreading to correct the grammar errors and typos, such as:
1) Line 1 of page 7: moreover -> Moreover?
2) Line 1 of Abstract: overtting -> overfitting?

In summary, the paper has to make significant improvements before it can meet the bar of ICLR.

---

### Official Review · AnonReviewer3 · 2018-11-09
**Recommend Reject as the contribution is incremental**

**Rating:** 3
**Confidence:** 5

**Review:**

This paper proposed to solve the instance-based transfer learning and feature-based transfer learning by stacking with a two-phase training strategy. The source data and target data are hybrid together first to train weak learners, and then the ensembled super learner is utilized to get the final prediction. Details for the stacking process are provided. Experimental results on MNIST-USPS, COIL, and Office-Caltech datasets show the proposed method can boost the performance, compared to TrAdaboost.

Pros:
The paper proposes to using stacking or ensembling to solve the domain adaptation problem, which shows some insight for further domain adaptation research.

Cons:
1. One of the main issues of this paper is the lack of novelty. The framework is incremented from the previous domain adaptation method such as TrAdaboost or BDA. For feature-based transfer learning, Equation (7)(8)(9) directly from the previous method.
2. Some arguments in this paper are not solid. For example, in the abstract,   the authors claim that under the two-stage training architecture, the fitting capability and generalization capability can be guaranteed at the same time. However, this is not well-justified in the following literature. Another example is "the settings of \lamda and N should be taken into consideration, if \lambda is too large, the performance of each learner can't be guaranteed, if \lambda is too small, training data can't be diversified enough" (page 7line 9~11)
3. This paper is weakened by the experimental part. Firstly, only TraDaboost method is used as a baseline. The paper can be largely improved by comparing with the state-of-the-art ensembling method for domain adaptation, for example:
Self-ensembling for visual domain adaptation, Geoff French,  ICLR 2018.
Secondly, the datasets used in this paper is small-scale and biased. It would be exciting to see how the proposed method will perform on the state-of-the-art large-scale domain adaptation dataset, for example, Office-Home dataset, Syn2Real dataset.

 Others:
1. Some terminologies used in this paper are confusing: (1) the h_t and c are not defined in Equation (2). in Algorithm 2, how to construct kernel matrix K_t using k_t?
2. The written of this paper can be largely improved. Some sentences are grammarly mistaken. Typos examples:
Abstract line 1: overtting -> overfitting
Section 2.1, we use TraAdaboost -> We use TrAdaboost
3. The citation style used in this paper is not correct.

Problems:
1. In section 2.2, what's the difference between the kernel matrix K with the unbiased estimate of MK-MMD (proposed by Gretton, NIPS 2012, also used in Deep adaptation network, Long, et al. ICML2015)?

---

### Meta-Review · Area_Chair1 · 2018-12-05
**Insufficient Novelty**

**Confidence:** 5
**Recommendation:** Reject

**Metareview:**

This work proposes a method for both instance and feature based transfer learning.
The reviewers agree that the approach in current form lacks sufficient technical novelty for publication. The paper would benefit from experiments on larger datasets and with more analysis into the different aspects of the proposed model.